# Species Diversity and Phylogenetic Relationships of Olive Lace Bugs (Hemiptera: Tingidae) Found in South Africa

**DOI:** 10.3390/insects12090830

**Published:** 2021-09-15

**Authors:** Vaylen Hlaka, Éric Guilbert, Samuel Jacobus Smit, Simon van Noort, Elleunorah Allsopp, Jethro Langley, Barbara van Asch

**Affiliations:** 1Department of Genetics, Stellenbosch University, Private Bag X1, Matieland 7602, South Africa; 23935731@sun.ac.za (V.H.); jethrolangley21@gmail.com (J.L.); 2Muséum National d’Histoire Naturelle, UMR 7179, CP50, 45 Rue Buffon, 75005 Paris, France; eric.guilbert@mnhn.fr; 3Centre for Novel Agricultural Products, Department of Biology, University of York, York YO10 5DD, UK; cobus.smit@york.ac.uk; 4Research and Exhibitions Department, Iziko South African Museum, P.O. Box 61, Cape Town 8000, South Africa; svannoort@iziko.org.za; 5Department of Biological Sciences, University of Cape Town, Rondebosch, Cape Town 7700, South Africa; 6Agricultural Research Council, Infruitec-Nietvoorbij, Private Bag X5026, Stellenbosch 7599, South Africa; AllsoppE@arc.agric.za

**Keywords:** *Cysteochila lineata*, DNA barcoding, *Neoplerochila paliatseasi*, *Olea europaea*, *Plerochila australis*

## Abstract

**Simple Summary:**

Olive lace bugs feed on wild and cultivated *Olea europaea*, causing a negative impact on plant vitality and development. These insects are known to affect olive orchards in South Africa, the country where most of the olive and olive products on the continent are produced. However, the diversity of species of these pests is not clear. Morphological analysis and DNA barcoding showed the presence of *Cysteochila lineata*, *Plerochila australis*, *Neoplerochila paliatseasi* and *Neoplerochila* sp. Further analyses of genetic divergence and phylogenetic clustering in 30 species in 18 genera of Tingidae using new and publicly available DNA barcodes showed that the majority of sequences deposited on BOLD Systems were correctly assigned to species. The complete mitochondrial genomes of the four species found in South Africa were sequenced to assess their phylogenetic position within Tingidae. The four olive lace bugs formed one cluster of species, and the genus *Cysteochila* was not monophyletic as *C. lineata* grouped with the other three olive lace bugs but *C. chiniana* was placed in a different cluster. This result suggests that lace bug species that feed on olive trees may have a common ancestor and calls for further research on potential adaptations to *O. europaea*.

**Abstract:**

Olive lace bugs (Hemiptera: Tingidae) are small sap-sucking insects that feed on wild and cultivated *Olea europaea*. The diversity of olive lace bug species in South Africa, the most important olive producer on the continent, has been incompletely surveyed. Adult specimens were collected in the Western Cape province for morphological and DNA-based species identification, and sequencing of complete mitogenomes. *Cysteochila lineata*, *Plerochila australis*, *Neoplerochila paliatseasi* and *Neoplerochila* sp. were found at 12 sites. Intra- and interspecific genetic divergences and phylogenetic clustering in 30 species in 18 genera of Tingidae using new and publicly available DNA barcodes showed high levels of congruity between taxonomic and genetic data. The phylogenetic position of the four species found in South Africa was inferred using new and available mitogenomes of Tingidae. Notably, olive lace bugs formed a cluster of closely related species. However, *Cysteochila* was non-monophyletic as *C. lineata* was recovered as a sister species to *P. australis* whereas *Cysteochila chiniana*, the other representative of the genus, was grouped with *Trachypeplus jacobsoni* and *Tingis cardui* in a different cluster. This result suggests that feeding on *O. europaea* may have a common origin in Tingidae and warrants future research on potential evolutionary adaptations of olive lace bugs to this plant host.

## 1. Introduction

Lace bugs (Hemiptera: Tingidae) comprise approximately 2500 species of small phytophagous insects in 300 genera distributed in all tropical and temperate continental and most oceanic regions except for the frigid zones [1]. Lace bug adults and nymphs feed by piercing the abaxial surface of the leaves of living plants to extract sap from cellular tissues [2]. Continuous feeding can result in chlorotic spots that may necrotize with detriment to plant vitality, and heavy infestations may cause premature death of young shoots and defoliation of the host. Lace bugs are generally monophagous, and a species feeds on the same kind of plant or group of closely related plants, including several agricultural crops and ornamentals.

Sub-Saharan Africa has a rich assemblage of native insects associated with Oleaceae, including several species of olive fruit flies and olive flea beetles, and a diversity of parasitoid, hyperparasitoid and olive seed wasps [3,4,5,6,7]. Lace bugs feeding on Oleaceae are only found in sub-Saharan Africa: *Catoplatus dilatatus* Jakovlev (on *Olea* sp.), *Cysteochila pallens* Horvath (on *O. chrysophylla*), *Cysteochila sordida* Stål (on *O. verrucosa*), *Olastrida oleae* Schouteden (on *O. europaea*) [8], *Cysteochila lineata* Duarte Rodrigues (on *O. capensis*), *Cysteochila nervosana* Drake and *Caffrocysta aliwalana* Duarte Rodrigues (on *Olea europaea* subsp. *cuspidata*) [9], and *Plerochila australis* (Distant) and *Neoplerochila paliatseasi* Duarte Rodrigues (on *O. europaea*) [10,11]. The exception to this pattern is *Froggattia olivinia* Froggatt, which is native to Australia and feeds not only on *Notelaea longifolia* (Oleaceae) but also on imported *O. europaea* [8].

In South Africa, cultivated olives are often grown in proximity to African wild olives (*Olea europaea* subsp. *cuspidata*), which may act as a source of both olive pests and their natural enemies. African wild olives and cultivated olives are closely related species; hence, most insects associated with African wild olives have been found to also occur on cultivated olives [5]. Despite the diversity of the native olive-associated entomofauna, South African olive growers face less aggressive threats from phytophagous insects, namely *Bactrocera oleae* Rossi (Diptera: Tephritidae), than their Mediterranean and Californian counterparts. Globally, South Africa is a small producer of boutique olive products mostly sold locally, but the industry also exports table olives and olive oil to neighbouring African countries, the European Union and the United States of America. Lace bugs affecting wild and cultivated olive trees in the Western Cape province, the most important region for olive production on the continent, are commonly referred to as “olive tingids” by local farmers and have been reported to include *P. australis* [12,13] and *N. paliatseasi* [11]. Perceptions on the extent of olive lace bug injury to cultivated olive trees vary from “olive tingids” being a minor pest that does not require management to a threat that impacts olive production and requires insecticide treatment.

The genus *Neoplerochila* was erected by Duarte Rodrigues [14] to hold the species *inflata* Duarte Rodrigues. *Neoplerochila* is only known from Namibia and South Africa and now includes eight species of which *N. millari* Göllner-Scheiding, *N. dispar* Duarte Rodrigues, *N. weenenana* (Drake) and *N. paliatseasi* are found on Oleaceae. The hosts of *N. inflata* Duarte Rodrigues, *N. katbergana* Drake, *N. uniformis* Duarte Rodrigues and *N. youngai* Duarte Rodrigues are presently unknown [10]. *Neoplerochila paliatseasi* is probably distributed countrywide in South Africa, as it was found in the Limpopo, North West and Western Cape provinces [10,11], and in Gauteng province in the present study.

The genus *Plerochila* was erected by Drake (1954) to hold *Plerochila australis* Distant and *P. horvathi* Schouteden as close to the genus *Cysteochila* but differing on the shape of paranota and carinae. *Plerochila* currently includes 17 species restricted to Africa, of which *P. australis*, *P. horvathi* and *P. rutshurica* Schouteden are known to feed on *Olea*. The host plants of most *Plerochila* species are not known [15]. *Plerochila australis* has been reported in Ethiopia, the Democratic Republic of Congo, Kenya, Madagascar, Mauritius, Mozambique, Namibia, Sudan, Tanzania, Uganda and South Africa [15]. In South Africa, *Plerochila australis* is probably distributed countrywide as it was reported in the Western Cape, Northern Cape, Gauteng, Limpopo, Mpumalanga and North West provinces [15].

The Mediterranean Basin and California are currently free from olive lace bugs, but these may become a threat if translocated from their original geographic range. Since our previous report of *N. paliatseasi* in South Africa [11], we learned that the species is indeed present on Madeira Island (Portugal), where it feeds on cultivated olives [16], but it seems to be restricted to that insular region and has not been found in mainland Europe, to the best of our knowledge. Under the right circumstances, olive lace bugs can become a problem, as is the case of the native Australian *F. olivinia*, which moved to cultivated olive trees and African wild olives when imported plants became established in the country starting in the 19th century [17]. *Froggattia olivinia* is now a serious pest of cultivated olives in New South Wales and Queensland, but it has not yet been reported outside Australia.

This work is part of a larger effort to catalogue the diversity of insect species associated with wild and cultivated olives in South Africa. The main objectives were (1) to gain further insights into the diversity of olive lace bugs found in South Africa and (2) to investigate the phylogenetic position of those species within the family Tingidae using new and publicly available mitogenome sequences.

## 2. Materials and Methods

### 2.1. Specimen Collection, Morphological Identification and DNA Extraction

Olive lace bugs were collected in nine areas in the Western Cape (cultivated and wild olive trees) and one site in Pretoria (wild olive trees) between November 2015 and March 2020 (Appendix A). Additionally, eight olive farms in the Western Cape were visited between October 2020 and March 2021 during the South African olive growing season when olive lace bugs are likely to be present. Specimens were collected from cultivated olive blocks identified as infested by farm workers. The number of trees surveyed at each farm varied from 10 to 50, and every second tree was sampled in any given block. Specimens were collected directly into individual plastic tubes, euthanized by freezing and stored in 100% ethanol at −20 °C until downstream analyses. DNA was extracted from individual specimens using a standard phenol-chloroform method [18] and stored at −20 °C until further use.

Morphological identification of ethanol-preserved specimens was performed by É. Guilbert following original descriptions, photos of type material and collections available [10,15]. Representative specimens of *C. lineata*, *P. australis*, *N. paliatseasi* and a nonspecific *Neoplerochila* were imaged and deposited in the entomological collection of the Iziko Museum (Cape Town) for future reference: *C. lineata* SAM-HEM-A012751, *Neoplerochila* sp. SAM-HEM-A012753, *P. australis* SAM-HEM-A010383 and *N. paliatseasi* SAM-HEM-A011647 (SAMC; Curator Simon van Noort). Codens of institutional depositories of voucher specimens follow Evenhuis (2019) [19]. Images were acquired with a Leica LAS 4.9 imaging system, comprised of a Leica^®^ Z16 microscope (using either a 2× or 5× objective) with a Leica DFC450 Camera and 0.63× video objective attached. The imaging process, using an automated Z-stepper, was managed using the Leica Application Suite V 4.9 software. Diffused lighting was achieved using a Leica LED5000 HDI dome.

### 2.2. DNA Barcoding

Specimens of *C. lineata* (*n* = 25), *P. australis* (*n* = 32), *N. paliatseasi* (*n* = 11) and *Neoplerochila* sp. (*n* = 14) were sequenced for the standard COI barcoding region (~650 bp) for assessing the congruency between morphological and DNA-based identifications using genetic clustering analysis and estimates of inter- and intraspecific genetic diversity. New species-specific PCR primers were designed for DNA barcoding of *P. australis* and *C. lineata*, based on their mitochondrial genomes (Appendix A). *Neoplerochila paliatseasi* and *Neoplerochila* sp. were barcoded using PCR primers specific to *N. paliatseasi* designed in a previous study [11]. Initial attempts to barcode *C. lineata* were made using the PCR primers specific to *P. australis* and the PCR primers specific to *N. paliatseasi*, but limited success (see Section 3.5) led to the design of species-specific primers for *C. lineata* once the mitogenome of latter species was assembled. All new species-specific PCR primer pairs anneal to the same COI region as the universal DNA barcoding primers, and were designed to be a perfect match to the COI sequence of each species. PCR amplifications were performed in a total volume of 5 μL containing 1× of KAPA2G Robust HotStart Ready Mix PCR kit (KAPPA Biosystems), 0.5 μM of each primer, 0.5 μL of MilliQ H2O, and 1.0 μL of template DNA (~100 ng), as follows: 95 °C for 3 min; 35 cycles of 95 °C for 15 s, 15 s at 58 °C for *C. lineata* and 54 °C for *N. paliatseasi* and *P. australis*, 72 °C for 1 min; and a final extension at 72 °C for 1 min. PCR products were sequenced using the reverse PCR primers specific to each species with the BigDye Terminator v3.1 Cycle Sequencing Kit (Applied Biosystems, Waltham, MA, USA) at the Central Analytical Facilities of Stellenbosch University, South Africa.

### 2.3. Intraspecific and Interspecific Genetic Diversity

All DNA barcodes assigned to Tingidae species were downloaded from the Barcode of Life Database (BOLD) Systems v4 (http://v3.boldsystems.org/, accessed on 20 October 2020) to provide a broader context for the intra- and interspecific divergence and genetic clustering patterns of the four olive lace bug species found in South Africa. The initial dataset included 1141 sequences that were subsequently filtered for (a) sequences identified to species level, (b) sequences with a minimum length of 500 bp overlapping the standard COI barcoding region, and (c) species represented by a minimum of three sequences. The final dataset included 367 sequences representing 30 species in 18 genera, including the new sequences (*n* = 82) generated in this study. Multiple sequence alignments were performed with the MAFFT algorithm [20] in Geneious Prime v2021.1 (https://www.geneious.com, accessed on 20 October 2020).

Genetic clustering of the COI sequence dataset of 30 species was assessed using a maximum likelihood (ML) tree constructed in IQ-Tree [21], with *Adelphocoris fasciaticollis* (NC_023796.1) (Hemiptera: Miridae) as outgroup. The best partitioning scheme was determined using the edge-linked greedy strategy [22] with automatic model selection [23,24] commands (−m MFP + MERGE). Branch supports were determined using 1000 replicates for both ultrafast bootstrapping and SH-aLRT branch tests [25,26]. The final ML tree was drawn using FigTree v1.4.4 (http://tree.bio.ed.ac.uk/, accessed on 4 September 2021).

Intra- and interspecific genetic divergences were estimated as p-distances (%) under the Kimura 2-parameter model [27] in MEGA X [28], with statistical support calculated from 1000 bootstrap replicates. Intraspecific diversity measures (number of haplotypes, number of polymorphic sites, haplotype diversity and nucleotide diversity) were calculated with Arlequin 3.5 [29]. Median-joining haplotype networks were constructed with Network 10.2, under the default settings [30]. The new sequences COI generated in this study were deposited on GenBank: *C. lineata* (MZ673445 to MZ673468), *N. paliatseasi* (MZ666853 to MZ666863), *Neoplerochila* sp. (MZ673417 to MZ673429) and *P. australis* (MZ676957 to MZ676987) (Appendix A).

### 2.4. Sequencing, Assembly and Annotation of Mitogenomes

The complete mitochondrial genomes for one specimen each of *C. lineata*, *P. australis* and *Neoplerochila* sp. were sequenced using the Ion Torrent™ S5™ platform (ThermoFisher Scientific, Waltham, MA, USA) available at the Central Analytical Facilities of Stellenbosch University, South Africa. Sequence libraries were prepared using the Ion Xpress™ Plus gDNA Fragment Library Kit (ThermoFisher Scientific, Waltham, MA, USA), according to the protocol MAN0009847 REV J.0. Libraries were pooled and sequenced using the Ion 540™ Chef Kit (ThermoFisher Scientific). The NGS reads of each species were mapped against the complete mitogenome of *N. paliatseasi* (MN794065) and assembled using Geneious Prime. Open reading frames were identified with Geneious Prime using the invertebrate mitochondrial genetic code. Transfer RNA genes (tRNAs) and their secondary structures were predicted using ARWEN software (http://130.235.244.92/ARWEN/, accessed on 15 March 2021) [31]. The two ribosomal RNA genes (12S rRNA and 16S rRNA) and the large non-coding region presumed to contain the control for transcription and translation (AT-rich region) were manually annotated by comparison with the mitogenomes of other Tingidae available on GenBank. The new complete mitogenomes of *C. lineata*, *Neoplerochila* sp. and *P. australis* were deposited on GenBank under the accession numbers MZ935684, MZ935685 and MZ935686.

### 2.5. Mitogenome Analyses

Nucleotide composition and compositional biases were calculated using Geneious Prime as AT skew = (A − T)/(A + T) and GC skew = (G − C)/(G + C)]. Relative synonymous codon usage was calculated in MEGA X. Repeated regions in the AT-rich region were identified using Tandem Repeats Finder v4.09 [32]. Start codons and overlapping and intergenic spaces were counted manually. Nonsynonymous (Ka) and synonymous (Ks) substitution rates were calculated using DnaSP6 [33].

### 2.6. Phylogenetic Reconstruction of Tingidae

The phylogenetic position of the olive lace bugs within Tingidae was assessed in the context of the 18 mitogenomes available for the family in GenBank as of October 2020, with *Apolygus lucorum* and *Adelphocoris fasciaticollis* (Hemiptera: Miridae) as outgroups (Appendix A). Individual PCGs were extracted from the complete mitogenome sequences and aligned using the translation algorithm in Geneious Prime. Stop codons were removed manually, and individual gene alignments were concatenated to form a single alignment. Poorly aligned regions and gaps in the concatenated alignment were eliminated using GBlocks v0.91b [34]. The final alignment was used to generate three sub-datasets: PCG123 (all codon positions), PCG12 (excluding the 3rd codon position), and an amino acid (AA) alignment.

Bayesian analyses were performed on the three datasets under the site-heterogeneous mixture model CAT-GTR in Phylobayes MPI in XSEDE v1.8c [35,36] to minimize the effect of mitochondrial compositional heterogeneity on phylogenetic reconstructions [37,38]. Constant sites were removed from the alignments and the minimum number of cycles was set to 30,000 with the burn-in set to 1000. The “maxdiff” was set to 0.3, and the minimum effective size was set to 50. Nodal support was estimated as Bayesian posterior probabilities (BPP). PhyloBayes analyses were run on the CIPRES Science Gateway Portal [39]. The final trees were drawn using FigTree v1.4.4.

## 3. Results and Discussion

Olive lace bug infestations are known to affect the development, health and fruit yield of cultivated olive trees in South Africa, but the diversity of the species present in the region has been incompletely described. A previous study reported the presence of *P. australis* and *N. paliatseasi* in the Western Cape on *Olea europaea*, but definite morphological identification and genetic data were only generated for *N. paliatseasi* at that point [11]. The present work study follows up by confirming the identity of *P. australis* and by reporting the presence of *C. lineata* and one *Neoplerochila* sp. genetically distinct from *N. paliatseasi*. Furthermore, the phylogenetic position of the four species was assessed in the context of mitogenomes publicly available for other Tingidae.

### 3.1. Morphological Identification of Olive Lace Bug Species

Images of representative specimens of *C. lineata*, *P. australis*, *N. paliatseasi* and *Neoplerochila* sp. analysed in this study are shown in Figure 1. *Cysteochila*, *Neoplerochila*, and *Plerochila* have a similar habitus with slight differences. All of the species analysed here have wide paranota reflexed onto the pronotum. The paranota are adjoined to the protunum in *C. lineata* and *P. australis* but not in the two *Neoplerochila* species. *Cysteochila lineata* has paranota reaching and partly covering the lateral carinae. *Plerochila australis* differs from the other species by the paranota being less developed and not reaching the lateral carinae. The paranota of both *Neoplerochila* species reach the lateral carinae but do not cover them. The costal area of these species is uniseriate, but *C. lineata* and *P. australis* have small and round areolae, while *N. paliatseasi* and *Neoplerochila* sp. have larger and subquadrate areolae. *Neoplerochila* sp. is very similar to *N. paliatseasi*: the only morphological difference would be the width of the sutural area, making the width of all the hemelytra quite uniform similar to *N. youngai,* and not narrowed opposite to the sutural area as in *N. paliatseasi*. Therefore, *N. paliatseasi* and *Neoplerochila* sp. could not be distinguished unambiguously, and comparative morphometric analyses for a full description of *Neoplerochila* sp. will have to be performed in the future.

### 3.2. Distribution of C. lineata, P. australis, N. paliatseasi and Neoplerochila sp.

Olive lace bug specimens were collected spanning a period of five years (2015–2020) over the course of other studies, but a survey of olive farms in the Western Cape was only performed during the South African olive-growing season of 2020. In total, 16 sites were visited, and specimens were collected from wild and cultivated ornamental trees in public and private spaces (Figure 2A). Most wild and cultivated trees from which specimens were collected showed typical symptoms of olive lace bug infestation, such as chlorotic spots and dried-out leaf tips (Figure 2C). Olive lace bugs were found in 12 sites out of the 16 sites visited (75%), including five out of the nine olive farms (56%) (Figure 3A). *Plerochila australis* was the most frequently found species (10 sites; 62.5%), *N. paliatseasi* and *C. lineata* were found at four sites (25%), and *Neoplerochila* sp. was only found at one site (Figure 3B). The four species were not found simultaneously at any site, but two sites had three species, two sites had two species, and 50% of the sites had only one species. Formal questionnaires were not performed, but some olive farmers mentioned using insecticides against olive lace bugs, in which case cultivated trees were sprayed twice a year. In cases of heavy infestations, insecticides have been used up to four times a year to significantly reduce populations. As insecticides represent additional economic and environmental costs, it will be interesting to investigate if olive lace bugs have efficient natural enemies that may contribute to manage infestations. In Australia, *F. olivinia* is reportedly difficult to manage and control, and low toxicity pyrethrum products are commonly used [17].

### 3.3. Haplotype Diversity of C. lineata, P. australis, N. paliatseasi and Neoplerochila sp.

*Neoplerochila* sp. was the least diverse of the four species as all specimens had the same haplotype. Haplotype diversity is the probability that two randomly selected haplotypes in the sample are different. *Cysteochila lineata*, which was found at four sites, had the highest haplotype diversity (H = 0.963), followed by *Plerochila australis*, the most frequently found species (H = 0.901) (Table 1). The haplotype diversity of *N. paliatseasi* (H = 0.787) was lower than that of *C. lineata* and *P. australis* but can still be considered high. Nucleotide diversity is the probability that two randomly chosen homologous nucleotide sites in the sample are different. The nucleotide diversity of the three species was relatively low, especially the of *N. paliatseasi* (π = 0.005); therefore, the two *Neoplerochila* were the least diverse species among the four olive lace bugs.

None of the species had high-frequency haplotypes or the classic star-like cluster around a central haplotype that is frequently interpreted as a sign of historical population expansion, and the intraspecific haplotype structure broadly reflected the diversity measures (Figure 4). However, the network of *C. lineata* showed several hypothetical haplotypes, and reticulations that were not present in the networks of the other two species. These features often result from the presence of multiple very low-frequency single nucleotide polymorphisms (SNPs) caused by erroneous base-calling due to the presence of double peaks. However, this was not the case as all sequences were of high quality and all SNP positions had single peaks in the electropherograms. It is possible that some of the sequences of *C. lineata* represent NUMTs despite the use of species-specific primers for PCR amplifications (see Section 3.5). If this is the case, the estimates of intraspecific genetic diversity presented here are inflated, but the broad genetic homogeneity and conspecificity of the specimens are not challenged.

### 3.4. Genetic Diversity of 30 Species of Tingidae Based on DNA Barcodes

The intra- and interspecific divergence of *C. lineata*, *Neoplerochila* sp. and *P. australis* were assessed in the context of DNA barcodes for other Tingidae retrieved from BOLD Systems. The genetic clustering analyses based on a ML tree showed that all species in the dataset formed monophyletic clusters with high statistical support (Figure 5). Intraspecific genetic divergence was estimated using maximum p-distances (Appendix A; Figure 6). Most species (83%) had intraspecific maximum p-distances below 2%, and only four species fell in the range between 2% and 3%, indicating a general trend of consistency in specimen identification. The only evident case of potential misidentification or cryptic diversity was *D. foliacea*, which had an intraspecific maximum p-distance of 9.28% due to the presence of a single highly diverged sequence (BOLD Record GMGMM1352-14) (Figure 5). Based on the results of a search on BOLD Systems using the “Identification” tool implemented on the platform, GMGMM1352-14 was most similar to a sequence deposited as *D. foliacea foliacea* collected in the Netherlands, but this sequence was not publicly available, as is the case for other *Derephysia*. Therefore, it was not possible to infer the monophyly of *Derephysia*. *Plerochila australis* had an intraspecific maximum p-distance of 3.29%, which may indicate ongoing differentiation in this group. The high divergence of *Neoplerochila paliatseasi* and *Neoplerochila* sp. as a single group supported the hypothesis of two distinct species (max p-distance = 7.12%). The choice of thresholds for intra- and interspecific distances is arbitrary, and no fixed value can be universally applied because variation in intraspecific divergence can be due to introgression, incomplete lineage sorting and recent speciation [40]. Nonetheless, estimates of sequence divergence can be useful for inferring patterns of genetic variation that allow for cataloguing of specimens and species into categories and complement morphological and ecological information when observable characters are absent, insufficient or non-informative. In the case of Tingidae, the data analysed here indicate that the range of maximum p-distances between 2% and 3% is a reasonable proxy for inferring conspecificity among sequences. Could this notion be extrapolated to congeneric pairs of species, i.e., are congeneric species of Tingidae consistently less diverged than non-congeneric species? Interspecific p-distances between all species pairs ranged from 5.22% to 28.72% (Appendix A). Among congeneric species pairs (*Acalypta*, *Corythucha*, *Gargaphia*, *Neoplerochila*, *Stephanitis* and *Tingis*), these values ranged from 5.21% to 22.83%, and among non-congeneric species pairs, the range was from 13.80% to 28.72%. The least diverged species pairs were indeed congeneric and involved most of the *Corythucha* species pairs, and the pair *N. paliatseasi*/*Neoplerochila* sp. (5.21–12.93%); however, the ranges of genetic divergences between congeneric and non-congeneric species pairs largely overlap, indicating that p-distances are not an adequate proxy for inferring congeners in Tingidae.

### 3.5. Potential Amplification of NUMTs in Cross-Species PCR

The assessments of genetic diversity and haplotype structure in *P. australis*, *C. lineata*, *N. paliatseasi* and *Neoplerochila* sp. were based on COI sequences obtained using newly designed species-specific PCR primers. However, we first attempted to barcode *C. lineata* using the primers specific to *P. australis* (Ple-F/Ple-R) and *N. paliatseasi* (Neo-F/Neo-R) because the mitogenome of *C. lineata* was the last to be sequenced in the course of this study. These cross-species amplifications of *C. lineata* generated sequences differing by several nucleotides for the same individual and nonspecific PCR products that resulted in low-quality sequences (data not shown). This may have been due to amplifications of nuclear mitochondrial DNA regions (NUMTs). Mitochondrial DNA is undoubtedly useful for assessments of genetic diversity due to its high copy number in the cell, lack of genetic recombination and high evolutionary rate [41], but the presence of NUMTs may confound the analyses. The loss of function of NUMTs results in the accumulation of sequence variation over time, which can be detected in protein-coding genes if amino acid codons are converted into internal stop codons and/or indels that generate frameshift mutations. These degraded versions are considered “older” (paleonumts) than “younger” NUMTs that are very similar to true mitochondrial DNA [42]. Spurious amplification of NUMTs may lead to overestimation of species diversity and erroneous conclusions in phylogenetic and phylogeographic studies. These problems have been pointed out as a major hindrance to DNA barcoding since the early stages of the development and dissemination of the methodology, which often relies on the use of “universal primers” for generating PCR amplicons across wide ranges of taxa [43]. In our study, the use of species-specific primers eliminated sequence inconsistencies in *C. lineata*, but this solution may not work across all insect taxa. In organisms with large genomes and that are, therefore, more tolerant to the presence of NUMTs, these are likely to be amplified even when species-specific primers are used. For example, a recent study showed that DNA barcoding in Orthoptera is particularly challenging due to the widespread presence of NUMTs, and other mitochondrial markers may have to be employed to infer species diversity correctly [44].

### 3.6. Mitogenomics of Olive Lace Bugs

The NGS runs generated 8,298,222 reads with an average length of 176 bp for *C. lineata*, 13,335,929 reads with an average length of 176 bp for *Neoplerochila* sp. and 15,637,818 reads with an average read length of 168 bp for *P. australis*. A total of 80,411 reads from *C. lineata*, 121,545 reads from *Neoplerochila* sp., and 43,401 reads from *P. australis* were mapped to the reference sequence (*N. paliatseasi*, NC_046031). The coverage of the final mitochondrial sequences of *C. lineata* (15,209 bp), *Neoplerochila* sp. (15,339 bp) and *P. australis* (15,208 bp) was 964×, 1384× and 473×, respectively. The quality of mitogenomes is directly associated with read coverage, and low coverage is known to result in sequence gaps and undetermined regions [45]. The read coverage of each final consensus sequence was high and largely exceeded the minimum required for mitogenomic studies (15×) [46]. The average length of the new mitogenomes (15,252 bp) was in line with other Tingidae (15,355 bp), and the three species had the typical set of mitochondrial genes found in Metazoa: 13 protein-coding genes (PCGs), 22 transfer RNA genes (tRNAs), two ribosomal (rRNAs) genes, and a non-coding region presumed to contain the control for replication and transcription (AT-rich region) (Figure 7). Twenty-three genes were located on the majority (J) strand, and 14 genes were located on the minority (N) strand (Appendix A). The gene arrangement of the PCGs, tRNAs, rRNA and putative control region (AT-rich region) was conserved in all Tingidae analysed here, and identical to the hypothetical Arthropoda ancestral [47].

The set of tRNA genes identified with ARWEN was manually compared with those identified in other Tingidae, and the most probable 22 tRNAs were annotated. All tRNAs had the typical cloverleaf-like structure, except tRNA^Ser1^ (TCT) of *C. lineata* and *P. australis* in that the dihydrouridine (DHU) arm was reduced and replaced with a simple loop, as commonly is the case in Metazoa [48] (Figure 8). In contrast, the DHU arm of tRNA^Ser1^ of *Neoplerochila* sp. and *N. paliatseasi* was complete. A complete DHU arm of tRNA^Ser1^ was also present in the mitogenomes of *Pseudacysta perseae* [49] and *Corythucha ciliata*, the latter of which had a different anticodon (TTC) [50]. The length of the tRNAs ranged from 63 bp (tRNA^Ala^) to 76 bp (tRNA^Thr^) in *C. lineata*, 61 bp (tRNA^Ala^) to 74 bp (tRNA^Lys^) in *Neoplerochila* sp., and 63 bp (tRNA^Cys^) to 75 bp (tRNA^Lys^) in *P. australis*. The location and average size of 16S rRNA (1228 bp; between tRNA^Leu1^ and tRNA^Val^) and 12S rRNA (771 bp; between tRNA^Val^ and the AT-rich region) in the four species were in line with the average size of the two genes in other Tingidae (1230 bp and 784 bp, respectively).

The large non-coding (AT-rich) region believed to contain the control for replication and transcription was located between 12S rRNA and the I-Q-M tRNA cluster, as in other Tingidae. The sizes of the AT-rich region were similar in the four olive lace bugs and ranged from 733 bp in *C. lineata* to 854 bp in *N. paliatseasi.* In other Tingidae, the AT-rich region ranged from 287 bp in *Tingis cardui* to 2215 bp in *Stephanitis chinensis*. Tandem repeats in the control region are common in animals, most likely as a result of slipped-strand mispairing during DNA replication [51]. *Neoplerochila* sp. had two repeats of a 170-bp motif at the 3′-end of the AT-rich region, which were separated by 4 bp and represented 40% of the region, *P. australis* had two repeats of a 166-bp motif separated by 6 bp representing 45% of the region, but no tandem repeats were identified in *C. lineata.* Tandem repeats in the 3′-end of the control region have also been identified in *P. perseae* (six repeats of 36 bp), *C. ciliata* (two repeats of 189 bp), and *N. paliatseasi* (two repeats of 156 bp) [11,49].

All PCGs in Tingidae started with a canonical ATN except for ND5 in *Corythucha marmorata* (GTG), and the most frequently used start codon was ATG (Figure 9). PCGs in *C. lineata*, *Neoplerochila* sp., *N. paliatseasi* and *P. australis* initiated with ATG in ATP6, COIII, COI, CYTB, ND1, ND4, and ND5; ATT in COII, ND2 and ND6; and ATA in ND4L and ND3, except for ND3 in *C. lineata* (GTG). The alternative start codon GTG has been previously found across a range of insect taxa including some species of Diptera [52], Mecoptera [53], Plecoptera [54], and Hemiptera such as *Sogatella furcifera* (Delphacidae) [55] and *Triatoma dimidiata* (Reduviidae) [56]. Most PCGs in the four olive lace bugs terminated with TAA, except ND4 in *Neoplerochila* sp., ATP6 and ND4 in *P. australis*, and CYTB in *N. paliatseasi*, which terminated with TAG. Incomplete stop codons (TA and T) were present in *C. lineata* (ND5, COII and ND2), *Neoplerochila* sp. (CYTB and ND5), *N. paliatseasi* (COII, ND3 and ND5) and *P. australis* (COII and ND5). Incomplete stop codons are common in animal mitochondrial genes and are presumed to be completed by posttranscriptional polyadenylation [57].

The mitogenomes of *C. lineata*, *Neoplerochila sp.* and *P. australis* were highly compact, with an average of 32 bp of intergenic nucleotides at 11 locations. The longest intergenic spacers were located between tRNA^Thr^ and COI in *C. lineata* (9 bp) and *Neoplerochila* sp. (12 bp) and between tRNA^Gln^ and tRNAMet in *P. australis* (10 bp), in line with *N. paliatseasi* and other Tingidae where the total number of intergenic nucleotides ranged from 1 to 48 bp. Olive lace bugs had an average of 19 gene overlaps, mostly involving tRNAs. The longest overlap (19 bp) was in *C. lineata* between ND4L and tRNA^Thr^, followed by 14 bp between COIII and ATP6 in *Neoplerochila* sp., 14 bp between ATP6 and COIII in *N. paliatseasi*, and 14 bp between tRNA^Gln^ and tRNA^Met^ in *P. australis*. The total number of gene overlaps varied in other Tingidae and was lower than the average for the four olive lace bugs, ranging from 8 bp in *P. perseae* to 17 bp in *C. ciliata*.

Olive lace bugs had the high A+T content typically found in insect mitogenomes, with an average of 75.1% for the total sequences. The A+T content of the AT-rich regions of the three new mitogenomes (*C. lineata*, 78.2%; *Neoplerochila* sp., 76.9%; *P. australis*, 75.2%) was higher than that of their complete sequences, which was also the case of *N. paliatseasi* and all other Tingidae except *C. ciliata* [49]. Olive lace bugs also had a similar A+T content for the combined tRNAs (average = 77.8%), and combined rRNAs (average = 79.2%). The A+T content of the total PCGs in olive lace bugs varied from 65.8% in *C. lineata* to 81.3% in *Neoplerochila* sp. and *N. paliatseasi*. The A+T content of individual PCGs was lowest in COI in all three species (66.4%), and highest in ATP8 (*C. lineata*, 79.5%) and ND4L (*P. australis*, 80.3%; *Neoplerochila* sp., 81.3%) (Appendix A).

The mitochondrial GC content varies among species and is influenced by mutation bias, selection and DNA repair bias on the complementary DNA strand [58]. According to the second parity rule, bases in the complementary DNA strand exist at equal frequencies when there are no mutations or selection bias [59]. The presence of AT and GC skews on the same DNA strand may indicate that the species underwent mutations or environmental selection [58]. *Cysteochila lineata*, *Neoplerochila* sp. and *P. australis* had positive AT skews and negative GC skews in most genes and AT-rich regions, except COI and CYTB. ND6 also did not have a positive AT skew and a negative GC skew in *Neoplerochila* sp. and *P. australis*. In the four olive lace bugs, four of 13 PCGs on the N strand had higher AT skews than PCGs on the J strand. The nucleotide bias towards A and T was reflected in codon usage, with AT-rich codons (UUU, UUA, AUU, AUA, UAU, AAU and AAA) representing an average of 42.3% of all codons. Relative synonymous codon usage (RSCU) for each codon is calculated as the relative frequency of a codon within a mitogenome. An RSCU value higher than 1.0 indicates an over-represented codon, whereas an RSCU value lower than 1.0 indicates an under-represented codon [60]. RSCU was higher than 1.0 in all synonymous codons, indicating that AT-rich codons are favoured (Appendix A).

The ratio of non-synonymous to synonymous nucleotide substitutions (Ka/Ks) is generally used as an indicator of selective pressure on protein-coding sequences among different species. A Ka/Ks ratio greater than 1 indicates positive selection, which is assumed to have occurred during the evolution of the sequence. Average Ka/Ks were calculated for individual PCGs across the 18 Tingidae species included in this study (Figure 10). All genes had Ka/Ks < 1, which indicates purifying or stabilizing selection, of which ATP8 had the highest Ka/Ks (0.65), and ND1, COI, COII, COIII and CYTB had the lowest Ka/Ks (0.20).

### 3.7. Phylogenetic Position of Olive Lace Bugs within Tingidae

The mitochondrial phylogeny of Tingidae has been recovered inconsistently across different studies, and this instability may be due to atypical sequence heterogeneity and high levels of mutation rates in the family and differences in phylogenetic methodological approaches [49,50,61,62,63,64]. Furthermore, patterns of non-monophyly in mitochondrial phylogenies can result from hybridization and introgression events, and incomplete lineage sorting during speciation, which are more likely to occur among recently diverged species than in older lineages species [65]. The phylogenetic positions of *C. lineata*, *N. paliatseasi*, *Neoplerochila* sp. and *P. australis* were recovered using the novel sequences and all Tingidae mitogenomes available on GenBank at the time of this study, except *Eteoneus sigillatus* (KU896784; unverified sequence). Phylogenetic analyses were restricted to PCGs because these have the advantage of being translatable and do not generally contain many length-variable regions at the genera and species level, and the third codon position is mostly neutral and not constrained by selection [66].

The PCG123 and PCG12 trees recovered the same topology with three unresolved nodes, but PCG12 had slighter higher support for some nodes (Figure 11A). The AA tree had only one unresolved node and high statistical support for most nodes (Figure 11B). All trees recovered *Phatnoma laciniatum* (Phatnomini) as basal to Tingini, and the same main clades but a different order of deeper nodes. The genera *Stephanitis* and *Corythucha*, which are represented by more than one species, formed monophyletic clades in agreement with previous phylogenies [11,62]. *Ammianus toi* and *Perissonemia borneenis* were also recovered as sister taxa in agreement with previous reconstructions [11]. The genus *Cysteochila* is represented by two species; however, these did not form a monophyletic clade as *C. lineata* was recovered as a sister species to *P. australis* but *C. chiniana* was placed in a different cluster with *T. jacobsoni* and *T. cardui*. Nevertheless, the four olive lace bugs were placed in the same phylogenetic cluster with high support in both trees. *Cysteochila* is a large genus holding around 136 species closely related to *Plerochila* in which species of *Cysteochila* have been transferred (*C. horvathi* Schouteden, and *C. tzitikamana* Drake). In fact, these genera need to be revised, as the morphological characters used to distinguish species show very small differences and many are homoplastic (see [1]). This is also the case for *Neoplerochila*, in which species of *Physatocheila* were transferred (*N. katbergana* Drake and *N. weenenana* Drake).

Our results show that *C. lineata*, *P. australis*, *N. paliatseasi* and *Neoplerochila* sp. share a mitochondrial ancestor and suggest that feeding on *O. europaea* may have a common evolutionary origin in lace bugs. To test this hypothesis, it will be necessary to confirm which other lace bug species feed on *O. europaea* and to determine their phylogenetic position within a wider range of Tingidae. Candidate species could include species that have been found on *Olea* such as *Neoplerochila millari*; *N. dispar*; *N. weenenana*; *P. horvathi*; *Cysteochila impressa* Horvath; *Physatocheila namibiana* Duarte Rodrigues; and the Australian *F. olivinia*, a known pest of cultivated olives.

## 4. Conclusions

Cultivated olive trees were introduced to the Western Cape province of South Africa less than 100 years ago, but African wild olives are widely distributed in this and other regions of South Africa. Previous works have shown that the entomofauna affecting *O. europaea* in sub-Saharan Africa most likely co-evolved with African wild olives. We confirm the presence of four species of olive lace bugs in South Africa (*C. lineata*, *P. australis*, *Neoplerochila* sp. and *N. paliatseasi*), of which *P. australis* was the most frequent. The four olive lace bugs have a close phylogenetic relationship among Tingidae, in agreement with their utilization of *O. europaea*. As relatively few olive lace bug species are adapted to *O. europaea*, it will be interesting to gather further evidence for a common origin of this feeding habit.

## Figures and Tables

**Figure 1 insects-12-00830-f001:**
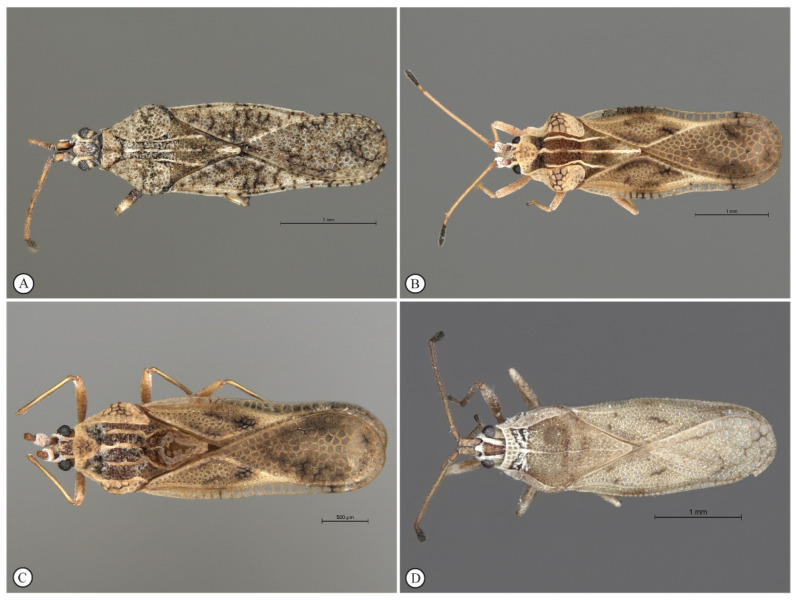
Representative adult specimens of olive lace bug (Hemiptera: Tingidae) species found in South Africa. (**A**) *Cysteochila lineata* Duarte Rodrigues (SAM-HEM-A01275), (**B**) *Neoplerochila paliatseasi* Duarte Rodrigues (SAM-HEM-A011647), (**C**) *Neoplerochila* sp. (SAM-HEM-A012753), and (**D**) *Pl**erochila australis* (Distant) (SAM-HEM-A010383).

**Figure 2 insects-12-00830-f002:**
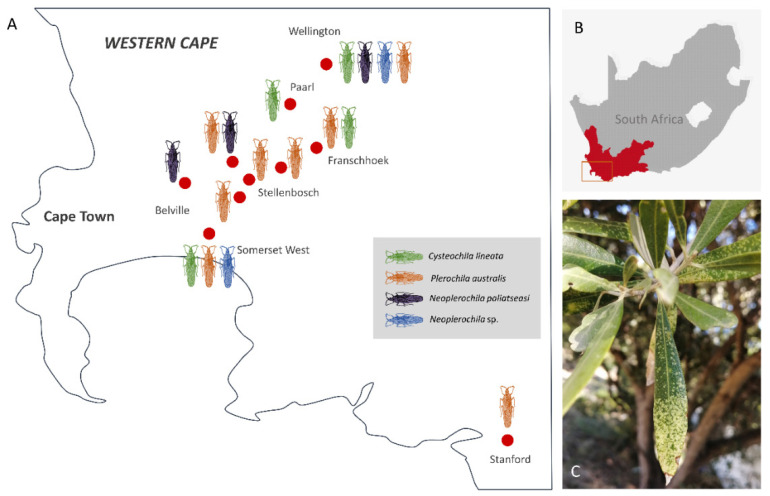
(**A**) Approximate geographic location of broad sampling areas (red dots) of olive lace bugs (*Cysteochila lineata*, *Neoplerochila paliatseasi*, *Neoplerochila* sp. and *Plerochila australis*) in the Western Cape province of South Africa. (**B**) Study area in the Western Cape. (**C**) Characteristic chlorotic spots on the leaves of a cultivated olive tree caused by feeding activity of olive lace bugs.

**Figure 3 insects-12-00830-f003:**
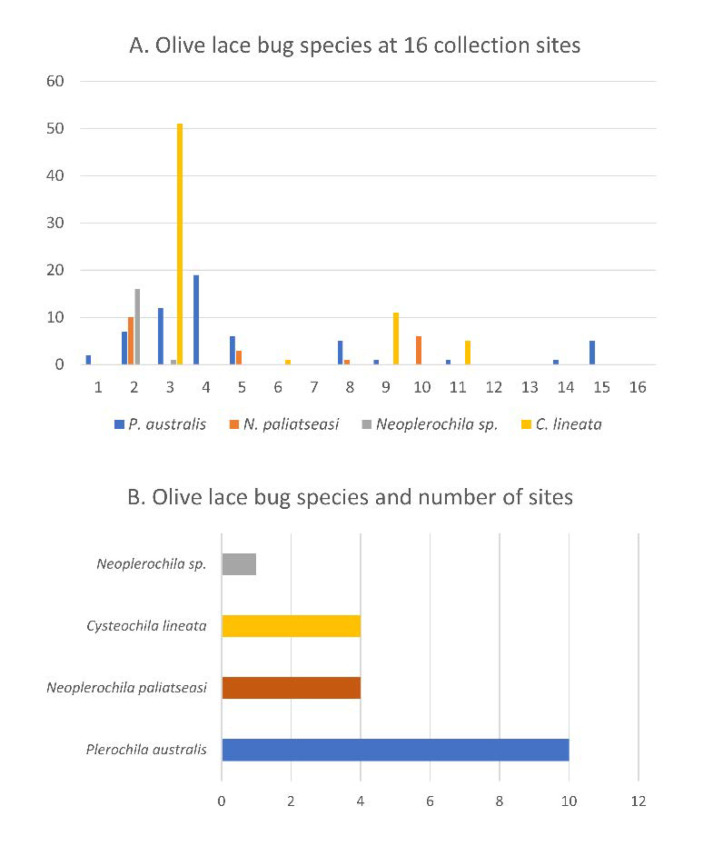
(**A**) Olive lace bug species found at 16 sample collection sites in the Western Cape province of South Africa. (**B**) Olive lace bug species and number of sites where each species was found.

**Figure 4 insects-12-00830-f004:**
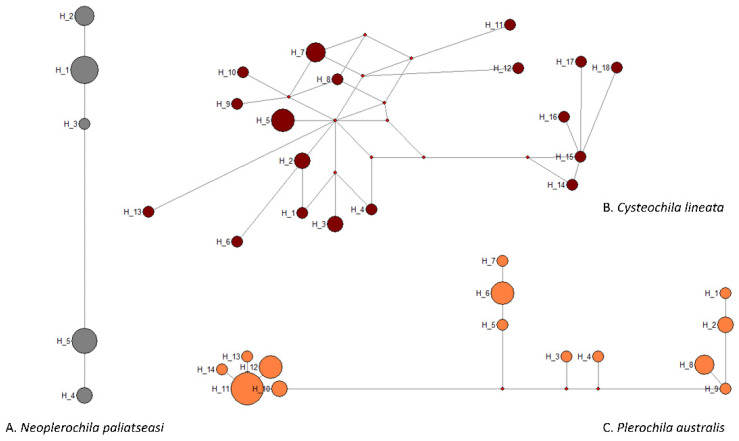
Median-joining network of COI haplotypes of three olive lace bug species (Hemiptera: Tingidae) found in South Africa.

**Figure 5 insects-12-00830-f005:**
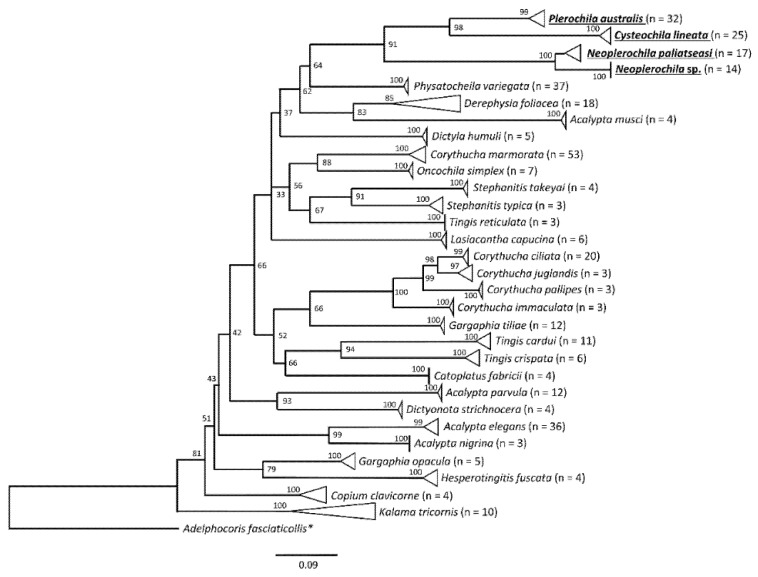
Maximum likelihood tree of lace bug species (Hemiptera: Tingidae) based on a 501-bp alignment of standard COI barcoding sequences. The analyses included 349 sequences representing 30 species in 18 genera retrieved from BOLD Systems and the new sequences of the olive lace bugs *Cysteochila lineata**, Neoplerochila*
*paliatseasi*, *Neoplerochila* sp., and *Plerochila australis* generated in this study (bold, underlined). Triangles represent collapsed groups of sequences belonging to the same species. * Outgroup (Hemiptera: Miridae).

**Figure 6 insects-12-00830-f006:**
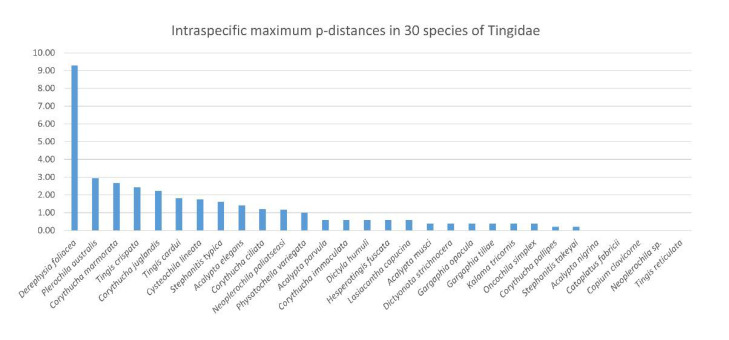
Intraspecific maximum p-distances (%) in 30 species of lace bugs (Hemiptera: Tingidae) based on a 501-bp alignment of the standard COI barcoding region (*n* = 349).

**Figure 7 insects-12-00830-f007:**
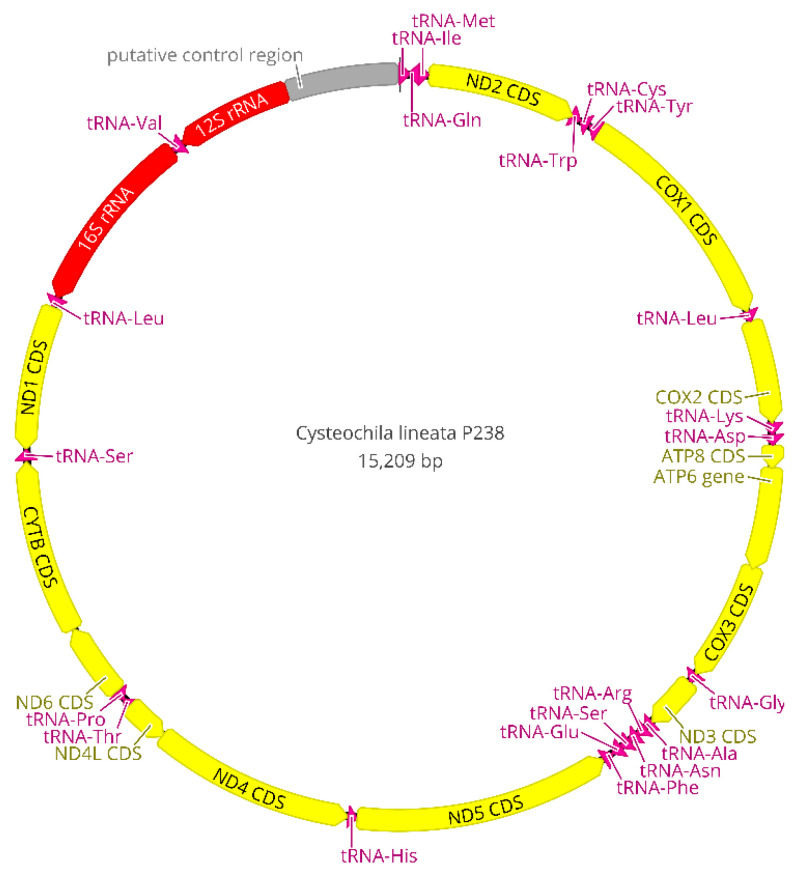
Circular map of the complete mitogenome of *Cysteochila lineata* (Hemiptera: Tingidae). Mitochondrial gene content and arrangement are conserved in Tingidae and identical to the hypothetical Arthropoda ancestor. Arrows indicate the direction of gene transcription.

**Figure 8 insects-12-00830-f008:**
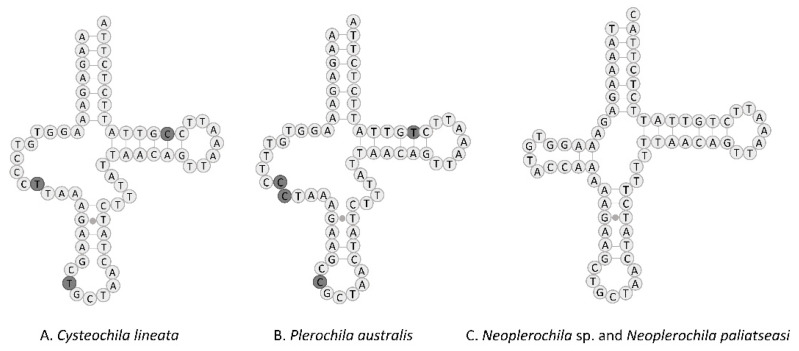
Predicted structure of tRNA^Ser1^ in the complete mitochondrial genomes of four olive lace bugs (Hemiptera: Tingidae) found in South Africa, with nucleotide differences highlighted. Inferred canonical Watson–Crick bonds are represented by lines, and non-canonical bonds are represented by dots.

**Figure 9 insects-12-00830-f009:**
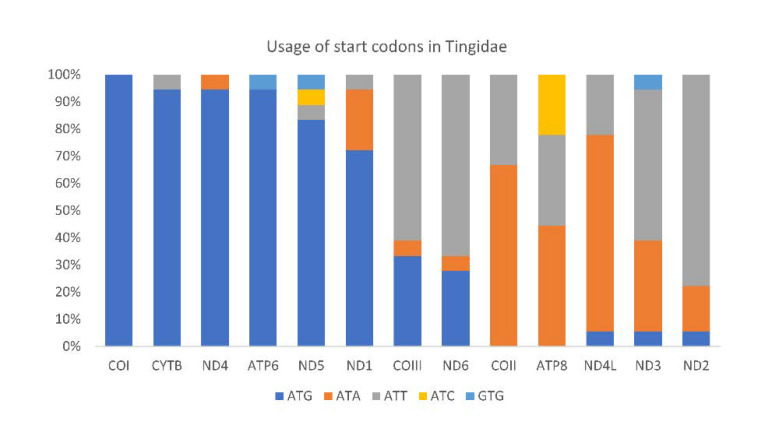
Usage of start codons in the complete set of 13 mitochondrial protein-coding genes in 18 species in the family Tingidae (Hemiptera).

**Figure 10 insects-12-00830-f010:**
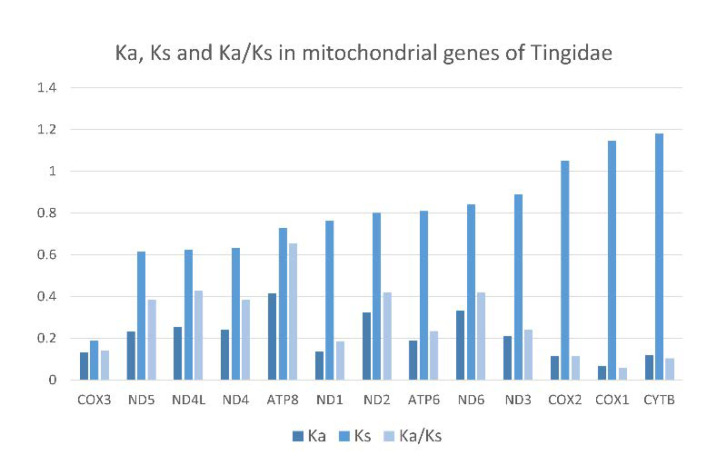
Evolutionary rates in 13 mitochondrial protein-coding genes of 18 species of Tingidae (Hemiptera). Ka—number of nonsynonymous substitutions. Ks—number of synonymous substitutions. Ka/Ks—ratio of the number of nonsynonymous to the number of synonymous substitutions.

**Figure 11 insects-12-00830-f011:**
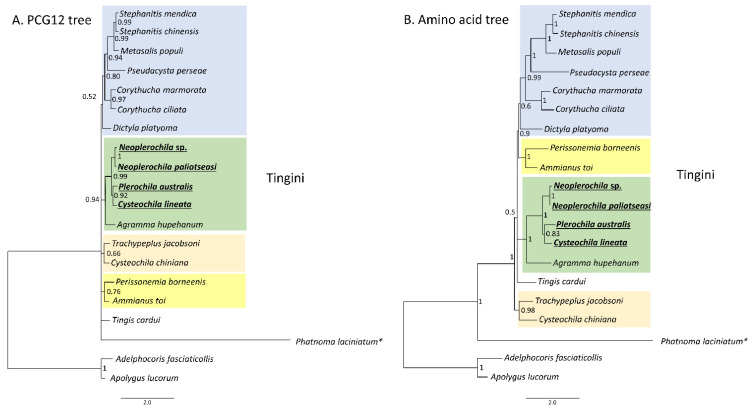
Phylogenetic relationships among 18 lace bug species (Hemiptera: Tingidae) based on 13 mitochondrial protein-coding genes. (**A**) PCG12-only first and second codon positions. (**B**) Amino acid tree. *Adelphocoris fasciaticollis* and *Apolygus lucorum* (Hemiptera: Miridae) were used as outgroups. Nodal statistical support is given as Bayesian posterior probability. * Phatnomatini.

**Table 1 insects-12-00830-t001:** Genetic diversity estimates for four species of olive lace bugs (Hemiptera: Tingidae) based on the standard COI barcoding region. k—number of haplotypes; S—number of polymorphic sites.

Species	n	k	S	Haplotype Diversity ± SD	Nucleotide Diversity ± SD
*Cysteochila lineata*	25	18	26	0.9633 ± 0.0235	0.007644 ± 0.004294
*Neoplerochila paliatseasi*	17	5	7	0.7868 ± 0.0590	0.005311 ± 0.003243
*Neoplerochila* sp.	14	1	0	n.a.	n.a.
*Plerochila australis*	32	14	25	0.9012 ± 0.0324	0.014416 ± 0.007612

## Data Availability

The DNA sequences generated in this study were deposited in GenBank under the following accession numbers: *C. lineata* (MZ673445 to MZ673468), *N. paliatseasi* (MZ666853 to MZ666863), *Neoplerochila* sp. (MZ673417 to MZ673429) and *P. australis* (MZ676957 to MZ676987).

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
