# Peer review of "Species Diversity and Phylogenetic Relationships of Olive Lace Bugs (Hemiptera: Tingidae) Found in South Africa"

_insects, 2021, doi:10.3390/insects12090830_

Round 1

Reviewer 1 Report

Lines 87-104: Ref 15 is not included in the text. Add it or delete it from the final chapter and re-order the rest bibliography numbers

Line 139-142: Which are the keys that were used for the morphological identification. Give some reference

Line 168-170: Refer to the different volumes and/or concentration analytically, eg “Χ μL of 0.5 μM of each primer”

Line 245-: italics, “… P. australis and N. paliatseasi…”

Line 257: insert comma – “…and P. australis, but not in…”

Figure 2A: Use different color for C. lineata, eg green, as it is not clearly differs from the N. paliatseasi.

Author Response

REVIEWER 1

Lines 87-104: Ref 15 is not included in the text. Add it or delete it from the final chapter and re-order the rest bibliography numbers

Reference was added.

Line 139-142: Which are the keys that were used for the morphological identification. Give some reference

References were added.

Line 168-170: Refer to the different volumes and/or concentration analytically, eg “Χ μL of 0.5 μM of each primer”

Done.

Line 245-: italics, “… P. australis and N. paliatseasi…”

Done.

Line 257: insert comma – “…and P. australis, but not in…”

Done.

Figure 2A: Use different color for C. lineata, eg green, as it is not clearly differs from the N. paliatseasi.

Done.

Reviewer 2 Report

The authors present novel data about the olive lace bugs collected from the Western Cape province for morphological and DNA-based species identification, and more importantly, they sequence and present four complete mitogenomes, based on which their phylogenetic positions were assessed. Interestingly, they recovered the relationships of Tingidae based on mitochondrial genomes and they recovered the non-monophyly of the genus Cysteochila. This work definitely contains new data, but the phylogenetic analyses appear to be out-of-data and require reconsideration 
(1) Mitogenome compositional heterogeneity: Why were the nucleotide dataset analyzed only under a site-homogeneous model? The PhyloBayes software, which allows different substitution processes for amino acid replacement at various sites, produced a tree that best matched known higher-level taxa and defined basal relationships in Hemiptera (refs. 60 and 61 in the MS) and Coleoptera (see Timmermans et al. 2015). Timmermans et al. (2015) also showed that the compositional heterogeneity cannot be eliminated for some mitochondrial genes, but dense taxon sampling and the use of appropriate Bayesian analyses (CAT-GTR+G model) can still produce robust phylogenetic trees. To infer a phylogenetic tree based on mitogenomic data, it is always better to use the CAT-GTR model accounting for compositional heterogeneity, as this phenomenon has been observed not only in Coleoptera (Cai et al., 2020), but also Neuroptera and Hemiptera. As such, the authors need to do extra runs of their data sets (cd12 and cd123) using the CAT-GTR model in PhyloBayes (can be also done in CIPRES).  
(2) It is not clearly indicated in the manuscript why NT datasets were used, but not AA dataset? Please also provide a tree using amino acid data, as it has been proven that AA (with 20 character states, instead of 4 in NT dataset) is more conservative and useful for inferring relationships of deeper nodes.

Please also consider citing the following papers:  
Cai, C., Tihelka, E., Pisani, D., & Donoghue, P. C. (2020). Data curation and modeling of compositional heterogeneity in insect phylogenomics: a case study of the phylogeny of Dytiscoidea (Coleoptera: Adephaga). Molecular Phylogenetics and Evolution, 106782.
Timmermans, M. J., Barton, C., Haran, J., Ahrens, D., Culverwell, C. L., Ollikainen, A., ... & Vogler, A. P. (2016). Family-level sampling of mitochondrial genomes in Coleoptera: compositional heterogeneity and phylogenetics. Genome Biology and Evolution, 8(1), 161-175.

Smaller points:
Neighbor-Joining tree of lace bug species (Hemiptera: Tingidae): this Neighbor-Joining method is totally out of date, so please use model-based phylogenetic analysis of the COI gene.

Line 177: what is BOLD Systems short for? Please add: The Barcode of Life Data System

In Figure 1: Add authorities for each species photographed. The authorities need to be added when a taxon (genus or species) first appears 

Author Response

REVIEWER 2

The authors present novel data about the olive lace bugs collected from the Western Cape province for morphological and DNA-based species identification, and more importantly, they sequence and present four complete mitogenomes, based on which their phylogenetic positions were assessed. Interestingly, they recovered the relationships of Tingidae based on mitochondrial genomes and they recovered the non-monophyly of the genus Cysteochila. This work definitely contains new data, but the phylogenetic analyses appear to be out-of-data and require reconsideration 

(1) Mitogenome compositional heterogeneity: Why were the nucleotide dataset analyzed only under a site-homogeneous model? The PhyloBayes software, which allows different substitution processes for amino acid replacement at various sites, produced a tree that best matched known higher-level taxa and defined basal relationships in Hemiptera (refs. 60 and 61 in the MS) and Coleoptera (see Timmermans et al. 2015). Timmermans et al. (2015) also showed that the compositional heterogeneity cannot be eliminated for some mitochondrial genes, but dense taxon sampling and the use of appropriate Bayesian analyses (CAT-GTR+G model) can still produce robust phylogenetic trees.

To infer a phylogenetic tree based on mitogenomic data, it is always better to use the CAT-GTR model accounting for compositional heterogeneity, as this phenomenon has been observed not only in Coleoptera (Cai et al., 2020), but also Neuroptera and Hemiptera. As such, the authors need to do extra runs of their data sets (cd12 and cd123) using the CAT-GTR model in PhyloBayes (can be also done in CIPRES).  

Agreed. The MrBayes trees were replace with PCG123, PCG12 and AA CAT-GTR trees constructed in PhyloBayes.

(2) It is not clearly indicated in the manuscript why NT datasets were used, but not AA dataset? Please also provide a tree using amino acid data, as it has been proven that AA (with 20 character states, instead of 4 in NT dataset) is more conservative and useful for inferring relationships of deeper nodes.

Agreed. An AA tree was added to the revised manuscript.

Please also consider citing the following papers: Cai, C., Tihelka, E., Pisani, D., & Donoghue, P. C. (2020). Data curation and modeling of compositional heterogeneity in insect phylogenomics: a case study of the phylogeny of Dytiscoidea (Coleoptera: Adephaga). Molecular Phylogenetics and Evolution, 106782. Timmermans, M. J., Barton, C., Haran, J., Ahrens, D., Culverwell, C. L., Ollikainen, A., ... & Vogler, A. P. (2016). Family-level sampling of mitochondrial genomes in Coleoptera: compositional heterogeneity and phylogenetics. Genome Biology and Evolution, 8(1), 161-175.

The suggested papers were consulted and cited as support for the use of CAT-GTR to minimize the effect of compositional heterogeneity on mitochondrial-based phylogenetic reconstruction.

Smaller points:
Neighbor-Joining tree of lace bug species (Hemiptera: Tingidae): this Neighbor-Joining method is totally out of date, so please use model-based phylogenetic analysis of the COI gene.

We replaced the NJ tree with an ML tree (IQ-tree). The results were identical to those obtained using the NJ tree.

Line 177: what is BOLD Systems short for? Please add: The Barcode of Life Data System

Added.

In Figure 1: Add authorities for each species photographed. The authorities need to be added when a taxon (genus or species) first appears 

Done.

Reviewer 3 Report

Authors of this manuscript use mitochondrial COI sequences generated in their facilities and those from the barcode of life database to investigate phylogenetic relationships among a group of lace bugs infesting olive species in South Africa. Authors use established standard techniques and procedures to generate and analyze data. 

Introduction seems to be too verbose and could be made shorter. In addition, the section on NUMTs could moved t discussion and made much shorter as NUMTs there are a lot of publications available (e.g. Hazkani-Covo, E., Zeller, R.M. and Martin, W., 2010. Molecular poltergeists: mitochondrial DNA copies (numts) in sequenced nuclear genomes. PLoS genetics6(2), p.e1000834.;  Richly, E. and Leister, D., 2004. NUMTs in sequenced eukaryotic genomes. Molecular biology and evolution21(6), pp.1081-1084.; Wang JX, Liu J, Miao YH, Huang DW, Xiao JH. Tracking the distribution and burst of nuclear mitochondrial DNA sequences (NUMTs) in Fig Wasp Genomes. Insects. 2020 Oct;11(10):680.)

Specific comments: 

Line 129: delete "haphazardly".

Lines 131 to 134: Too much details.  Just state that insects were collected in eight olive farms in Western Cape between Nov.  2015 and March 2020. 

Line 164: Italicize C. lineata.

Figure 6: Table 2 and Fig. 6 offer the same information. Use one or the other.

Line 439:  What reference sequence.  Please indicate this is mtDNA sequence of N. paliatseasi.

Author Response

REVIEWER 3

Authors of this manuscript use mitochondrial COI sequences generated in their facilities and those from the barcode of life database to investigate phylogenetic relationships among a group of lace bugs infesting olive species in South Africa. Authors use established standard techniques and procedures to generate and analyze data. 

Introduction seems to be too verbose and could be made shorter.

We prefer to decline this suggestion because information is not given in a redundant manner, and the present study is one of the few to address the knowledge gap on olive lace bugs. However, we did simplify some phrases and eliminated some less important information to decrease the total word count.

In addition, the section on NUMTs could moved t discussion and made much shorter as NUMTs there are a lot of publications available (e.g. Hazkani-Covo, E., Zeller, R.M. and Martin, W., 2010. Molecular poltergeists: mitochondrial DNA copies (numts) in sequenced nuclear genomes. PLoS genetics6(2), p.e1000834.;  Richly, E. and Leister, D., 2004. NUMTs in sequenced eukaryotic genomes. Molecular biology and evolution21(6), pp.1081-1084.; Wang JX, Liu J, Miao YH, Huang DW, Xiao JH. Tracking the distribution and burst of nuclear mitochondrial DNA sequences (NUMTs) in Fig Wasp Genomes. Insects. 2020 Oct;11(10):680.)

Unfortunately, the NUMT section cannot be moved because this paper presents the results and discussion in combined (single) section. However, we have shortened the NUMT section, following the suggestion.

Specific comments: 

Line 129: delete "haphazardly".

Deleted.

Lines 131 to 134: Too much details.  Just state that insects were collected in eight olive farms in Western Cape between Nov.  2015 and March 2020. 

We must decline this suggestion because it is not correct that we only collected specimens from eight olive farms between Nov 2015 and March 2020. In fact, non-farm locations were also visited (See Table S1). Moreover, the suggested simplification would result in the omission of information of relevance for the local olive industry.

Line 164: Italicize C. lineata.

Done.

Figure 6: Table 2 and Fig. 6 offer the same information. Use one or the other.

Table 2 was moved to Supplementary Material.

Line 439:  What reference sequence.  Please indicate this is mtDNA sequence of N. paliatseasi.

GenBank accession number was added.

Round 2

Reviewer 3 Report

Authors have sufficiently addressed the issues raised by the reviewers and I believe the manuscript is acceptable in current form subject to minor editorial adjustments.